# Homocysteine: Its Possible Emerging Role in At-Risk Population Groups

**DOI:** 10.3390/ijms21041421

**Published:** 2020-02-20

**Authors:** Elena Azzini, Stefania Ruggeri, Angela Polito

**Affiliations:** CREA—Research Centre for Food and Nutrition Via Ardeatina, 546 Rome, Italy

**Keywords:** homocysteine, pregnancy, children, adolescents, elderly

## Abstract

Increased plasma homocysteine is a risk factor for several pathological disorders. The present review focused on the role of homocysteine (Hcy) in different population groups, especially in risk conditions (pregnancy, infancy, old age), and on its relevance as a marker or etiological factor of the diseases in these age groups, focusing on the nutritional treatment of elevated Hcy levels. In pregnancy, Hcy levels were investigated in relation to the increased risk of adverse pregnancy outcomes such as small size for gestational age at birth, preeclampsia, recurrent abortions, low birth weight, or intrauterine growth restriction. In pediatric populations, Hcy levels are important not only for cardiovascular disease, obesity, and renal disease, but the most interesting evidence concerns study of elevated levels of Hcy in autism spectrum disorder (ASD) and attention deficit hyperactivity disorder (ADHD). Finally, a focus on the principal pathologies of the elderly (cardiovascular and neurodegenerative disease, osteoporosis and physical function) is presented. The metabolism of Hcy is influenced by B vitamins, and Hcy-lowering vitamin treatments have been proposed. However, clinical trials have not reached a consensus about the effectiveness of vitamin supplementation on the reduction of Hcy levels and improvement of pathological condition, especially in elderly patients with overt pathologies, suggesting that other dietary and non-dietary factors are involved in high Hcy levels. The importance of novel experimental designs focusing on intra-individual variability as a complement to the typical case–control experimental designs and the study of interactions between different factors it should be emphasized.

## 1. Introduction

Homocysteine (Hcy) is a significant biomarker for overall health status and, although it is not clear whether the Hcy represents an indicator rather the etiology of disease, a direct relationship between its elevated fasting plasma levels and several pathological disorders, including bone health [1], neurodegenerative disease, [2], renal dysfunction [3], cognitive impairments [4], and congenital defect development [5], as well as its status of independent risk factor in coronary heart and cerebrovascular disease, is widely supported by scientific literature [6,7,8]. 

Hcy is a sulfur-containing amino acid not present in dietary protein or used for its endogenous synthesis, but represents an intermediate in methionine metabolism. Methionine is an essential sulfur-containing amino acid present in various protein foods such as meat, eggs, dairy products, and legumes. Figure 1 represents the pathways of Hcy metabolism, which include a system of transmethylation, remethylation, and transsulfuration paths. In most cells, by transmethylation route Hcy and methionine cycle metabolically, the methyl group on activated methionine (*S*-adenosyl-methionine or SAM) may be added to methyl acceptors (DNA, RNA, and protein) by methyltransferases, and the *S*-adenosyl-Hcy (SAH) is rapidly hydrolyzed to adenosine and Hcy, which could improve its concentration. Transmethylations, chemical reactions transferring a methyl group from one compound to another, are generally regulated by the intracellular concentration of involved compounds; thus, SAM and SAH concentrations determine a cell’s methylation balance. Once formed, Hcy can be recycled into methionine or converted into cysteine by remethylation and transsulfuration routes, respectively. Hcy is remethylated to methionine through two separate reactions catalyzed by three different enzymes. In all tissues, folic acid donates a methyl group across methylenetetrahydrofolate reductase (MTHFR) in a reaction catalyzed by methionine synthase, a vitamin-B12-dependent enzyme [9]. Otherwise, mainly in the human heart, liver, and kidneys, Hcy is remethylated using betaine, which donates a methyl group by betaine-Hcy *S*-methyltransferase (BHMT) by a route independent of the one-carbon metabolism. Betaine can be found in several dietary sources, including wheat germ or bran, spinach, beets, seafood, and legumes. Studies have confirmed betaine’s ability to reduce Hcy levels in the face of excess methionine intake, as well as the fact that low-dose betaine supplementation leads to immediate and long-term lowering of plasma Hcy in healthy men and women [10,11]. This remethylation process begins when there are low concentrations of Hcy and methionine [12]. Alternately, mainly in the liver, but also in the kidneys, small intestine, and pancreas [13], Hcy is enzymatically modified by cystathionine-β-synthase, a B6-dependent enzyme, to irreversibly form cysteine through the intermediate cystathionine. The transsulfuration route results in sulfur metabolites including GSH, a key cellular antioxidant, and hydrogen sulfide (H_2_S), acting like a gaseous signaling molecule. The transsulfuration path starts to function when the concentrations of Hcy and methionine increase, for example by post-prandial protein intake [14], or cysteine is needed.

In humans, during protein synthesis, Hcy instead of methionine could be wrongly selected in the reaction catalyzed by methionyl-tRNA synthetase (MetRS), producing a reactive toxic metabolite of Hcy known as Hcy-thiolactone [15,16]. This compound is able to chemically modify proteins and to impair their normal function, generating *N*-homocysteinylated (N-Hcy) proteins. This increasingly prominent protein modification, called *N-*homocysteinylation, seems to exert cytotoxic, pro-inflammatory, prothrombotic, and pro-atherogenic properties, contributing to the cardiovascular and neurologic disorders of hyperhomocysteinemia (HHcy) [17,18]. Perła-Kaján et al. [19] demonstrated that Hcy-thiolactone-hydrolyzing enzymes, known as the paraoxonase (*PON*) multigene family, associated in the bloodstream with high-density lipoproteinis (PON1/HDL) detoxify homocysteine thiolactone. Therefore, genetic variation in *PON1* could compromise this activity in humans. In serum or plasma, but not in urine, Hcy-thiolactone and Hcy can be interconverted by enzymatic and/or nonenzymatic reactions, indicating that interconversions do not contribute significantly to urinary Hcy-thiolactone or Hcy, while their presence is attributable to elimination or excretion from the blood. Hcy-thiolactone produced in the human body is efficiently cleared by the kidneys and excreted in the urine. The urinary Hcy-thiolactone levels are 100-fold higher than plasma levels. Approximate concentrations of Hcy-thiolactone (11–473.7 nmol/L; *n* = 19) were estimated in the urine of healthy volunteers with respect to plasma (<0.1–22.6 nmol/L; *n* = 20) [20]. The similar renal clearance of Hcy-thiolactone to that of creatinine indicates that Hcy-thiolactone is not reabsorbed in renal tubules; more than 95% is excreted in the urine. In contrast, 99% of Hcy is reabsorbed and 1% excreted in the urine [21]. Hcy-thiolactone can contribute significantly to Hcy pools, particularly in urine. As suggested by Jakubowsky [22], Hcy-thiolactone has adverse effects on physiological function, underscoring the importance of examining urinary and plasma Hcy-thiolactone in addition to Hcy in the context of human disease. Modification by Hcy-thiolactone seems to explain Hcy toxicity in autoimmune response, cellular toxicity, and atherosclerosis. This has been proposed as a mechanism in human subjects with HHcy caused by mutations in cystathionine β-synthase (*CBS*), the most common of the genetic causes of severe hyperhomocysteinemia, associated with increases up to 40-fold in fasting Hcy, and classic homocystinuria (congenital homocystinuria) or *MTHFR* genes [23] which has been associated with mild (13–24 µM) and moderate (25–60 µM) HHCy levels [9]. In neuronal degeneration and age-related neurodegenerative diseases, the Hcy neurotoxicity is due to troubled mechanisms of methylation and/or redox potentials triggering increased calcium influx [24], amyloid and tau protein accumulation [25,26], apoptosis, and neuronal death [27,28]. 

When Hcy metabolic pathways are deregulated, the intracellular Hcy enters the circulation, increasing plasma levels. At this stage, its fate is to bind to plasma proteins or to be eliminated via the renal route. Under normal conditions, Hcy is reabsorbed in the renal tubules and its urinary excretion is negligible, although there is no evidence that homocysteine is actively removed by the human kidney. The precise mechanism by which the glomerular filtration rate is related to plasma homocysteine concentration has not been definitively established, but there is a good degree of clinical evidence that HHcy does not cause renal insufficiency [29]. A comprehensive scheme representing Hcy transport across compartments was recently proposed by Hannibal and Blom [30]. At intracellular levels in its reducing condition, Hcy levels can be kept within a limited range of concentrations to enable adequate remethylation via methionine-synthase MS and to prevent accumulation of SAH through the concerted contributions of the methionine and folate cycles. The excess cellular Hcy (Hcy-SH) is exported into the bloodstream via an unknown cellular transporter(s). Several mechanisms have been proposed for the removal of Hcy to the extracellular compartment. It has been reported that an excess of Hcy in the extracellular compartment can be exported to the extracellular medium via transmembrane transport systems. Hcy shares carrier systems with cysteine in a preferable order of alanine–serine–cysteine (ASC) > aspartate and glutamate (XAG) = large branched-chain neutral amino acids (L) transporter systems in vascular endothelial cells and ASC > L > XAG in smooth muscle cells. The sodium-dependent ASC system plays a predominant role in Hcy transport in vascular cells. [31]. Moreover, three different amino acid transport systems, L, A, and y+L, have been identified for the transport of Hcy across the microvillous plasma membrane of the human placenta [32]. The specifity of Hcy transport is probably due to the ionic composition of the medium. In the more oxidizing environment found in plasma, homocysteine either reacts with protein Cys residues (R-SH) to homocysteinylate these targets (R-S-S-Hcy), or undergoes oxidization to form the disulfide homocystine (Hcy-Hcy) or mixed disulfides with cysteine (Hcy-Cys) and other low-molecular-weight thiols (Hcy-SR). Furthermore, the authors [30] hypothesized that homocystine and mixed disulfides derived from plasmatic Hcy oxidation can enter kidney and liver cells via another yet unidentified transporter, where the reducing environment of the cells can regenerate Hcy. Additionally, a proportion of the homocysteine is found in a protein-bound form, mainly bound to albumin, and enters the cells by endocytosis of these proteins. Sengupta et al. [33] proposed that homocysteine can be delivered to cells via its association with albumin, since albumin is known to bind to at least three endothelial cell membrane proteins, including transcytosed via plasmalemmal vesicles (gp 60), or endocytosed via receptor mediated endocytosis (gp30, gp18). Mudd et al. [34] described the Hcy and the major related disulfides present in normal human plasma. Figure 2 shows the several forms of Hcy present in human blood. The free sulfhydryl form, Hcy, is present in low amounts in blood (1%–2%) as a protein-bound mainly bound to albumin (about 70% to 85% in normal subjects) and soluble disulphides (30%–25%; e.g., homocystine or cysteine-SS-Hcy). The amount of total Hcy is the sum of these three components. Values obtained could vary by laboratory and collection method; some variable changes in Hcy have been observed post-prandially, so collection of fasting samples is recommended.

To date, there is no general agreement on Hcy cut-off values. Since there a variety of factors affect Hcy levels, several cut-off points have been proposed [35], taking into account the imbalance of Hcy homeostasis in determining reference values. Jacobsen identified approximately 5 μmol/L and 25 μmol/L as ideal levels for an age range from 0 to 100 years [36]. Hcy reference range values within 4.7–14.6 μmol/L and 18.8–49.7 μmol/L have been reported for fasting and 6 h post-methionine load, respectively [37].

Even if correct functioning of the enzymes involved in the metabolic pathways of Hcy has an essential role in determining the plasma and urine levels of Hcy, several modifiable factors exert a strong effect on its concentration, thus characterizing a wide range of pathologies. These factors include adequate dietary intake of folate and vitamins B6 or B12, intake of proteins rich in methionine that help to regulate Hcy biochemical pathways, and unhealthy lifestyle choices such as alcohol abuse, high coffee intake, smoking habits, and use of drugs that may interfere with vitamin utilization and thus could increase the risk of HHcy-related pathologies [38]. What this means is that various diseases may have different etiologies in their respective relationships to plasma Hcy [39]. Considering its pivotal role, the nutritional management of elevated Hcy levels could provide powerful protection against its toxicity and unhealthy effects on human wellbeing. Several studies [40] have assessed the effects of folic acid, vitamin B6, and vitamin B12 supplements on elevated Hcy levels. Reductions in plasma Hcy concentrations in the range of about 25% to 30% and 10% to 15% were achieved in populations without or with folic acid fortification, respectively, upon dietary supplementation with folic acid and B12. A large prospective study reported a significant lowering effect on Hcy levels of daily supplementation combining folic acid, vitamin B6, and vitamin B12 [41]. Daily folic acid supplementation with 0.5 to 5.0 mg lowers plasma Hcy levels by about 25%. Daily supplementation of at least 0.4 mg vitamin B12 further reduces levels by about 7%, while vitamin B6 supplements can improve Hcy levels after methionine loading [42,43]. However, dose amount, bioavailability, and supplement type should be taken into account, as well as being adjusted for age, sex, and treatment assignment in analyzing the possible outcomes.

Additionally, various polymorphisms of the genes coding for MS, MTHFR, and CBS causing the low or absent activity rates of these proteins involved in Hcy clearance have an effect on circulating Hcy levels [44,45,46,47]. Kluijtmans et al. [47] reported at least five common functional polymorphisms that impair the activity of enzymes involved Hcy metabolism, including methylenetetrahydrofolate reductase [MTHFR] 677C>T and 1298A>C, methionine synthase 2756A>G, cystathionine β-synthase [CBS] 844ins68, and methionine synthase reductase 66A>G. These genetic variations affecting Hcy concentrations seem to be more significant in early-life than middle-aged subjects; it is possible that control of dietary factors would reduce homocysteine levels.

As already underlined, overproduction of Hcy could be harmful and is a possible cofactor in the etiology of multifactorial diseases, and may be a biomarker of chronic and acute inflammatory status and diseases, increasing the risk of other diseases and complication. Several studies and literature reviews have examined the role of Hcy in pathological conditions in specific population groups [48].

The aim of the present review article was to conduct a critical literature examination on the features of Hcy metabolism in different population groups at varying life stages, especially in risk conditions (pregnancy, infancy, old age), and on its relevance as a marker or etiological factor of diseases in these age groups, focusing on the nutritional treatment of elevated Hcy levels. Even if our review could not be exhaustive, we attempted to sum up what we consider the current state of the art of the most relevant and significant works about the role of Hcy in more vulnerable groups of people.

## 2. Pregnancy

The significance of HHcy levels in pregnancy has been investigated in relation to the increased risk of adverse pregnancy outcomes such as small size for gestational age at birth, preeclampsia, recurrent abortions, low birth weight, intrauterine growth restriction, and neural tube defects (NTDs) [49,50,51].

One systematic review [52] reported a correlation between HHcy levels and different complications in pregnancy; in particular, it was underlined that preeclampsia and placental abruption may occur more often in pregnant women with serum Hcy levels from 9.0 to 15.0 micromoles). In a previous study, Dodds et al. [53] demonstrated no significant correlation between Hcy levels in early pregnancy and risk of small size for gestational age at birth; meanwhile, a positive correlation between Hcy concentration at the top decile and pregnancy loss and preeclampsia was found (RR 2.1, 95% CI 1.2–3.6 and RR 2.7, 95% CI 1.4–5.0, respectively). The study did not observe a dose–response relationship according to quartile of Hcy concentration, due to the small sample size.

In one prospective study [54], Hcy levels and their effects on pregnancy and neonatal outcomes, including gestational diabetes, preeclampsia, gestational age at delivery, preterm birth, small size for gestational age, neonatal birth weight, and congenital abnormalities, were assessed in each trimester in 278 pregnant Korean women. Hcy levels were not significantly different among any of the trimesters (4.4–38.0 µmol/L), as observed by Bondevik et al. [55] in a group of Nepali women. No significant association was observed in Korean women between Hcy and any of the maternal or neonatal outcomes examined, including gestational age at delivery and birth weight of babies.

Mascarenhas et al. [56] suggested that the evaluation of Hcy levels during pregnancy should be done between 8 and 12 weeks to give a good indicator for pregnancy outcome. Increased serum Hcy in the first trimester of pregnancy is associated with history of pregnancy losses, hypertensive disorders of pregnancy, and preterm birth. This is also associated with hypertensive disorders of pregnancy, pregnancy loss, oligohydramnios, meconium stained amniotic fluid, and low birth weight in the current pregnancy.

Hcy levels during pregnancy are influenced both by the vitamin B nutritional status of women and by some specific physiological factors including increased glomerular filtration, rate hemodilution, and endocrinological changes occurring in this period, leading under normal conditions to a reduction of Hcy in the mid-pregnancy period [57]. Moreover Gaiday et al. [52] concluded that Hcy levels during pregnancy seem to also be linked to the geographical, cultural, and social characteristics of populations.

In the largest study (R Generation Study) about the associations of homocysteine, folate, and vitamin B12 levels and pregnancy outcomes, Bergen et al. [58] recruited 5805 women at gestational stage of 13.4 weeks (range: 11.4–16.5 weeks), and found a positive correlation between HHcy (highest quintile = 8.3 μmol/L) and birth weight (difference 110 g; *p* < 0.001), lower placental weight (difference 30 g; *p* < 0.001), and increased risk of small size for gestational age at birth (odds ratio (OR) 1.7; *p* = 0.006) in the lowest quintile (=5.8 μmol/L). Folate concentrations (lowest quintile = 9.2 nmol/L) were associated with lower placental weight (difference 26 g; *p* = 0.001) and birth weight (difference 125 g; *p* < 0.001), and increased risks of small size for gestational age at birth (OR 1.9; *p* = 0.002), prematurity (OR 2.2; *p* = 0.002), and preeclampsia (OR 2.1; *p* = 0.04) compared with the highest quintile (25.9 nmol/L). Interesting data have been recorded regarding folic acid supplementation: 18.8% of pregnant women (*n* = 1.091) did not use any folic acid supplements, 33.1% of women (*n* = 1.921) started folic acid supplementation in the pre-conceptional period, 24.5% (*n* = 1.422) started before 8 weeks of pregnancy, and data were missing for 23.7%.

As regards neural tube defects, low vitamin B12 and B9 status are risk factors for neural tube defects (NTDs) as they lead to an increase of Hcy levels [59,60,61]. The monitoring of Hcy levels might be important in understanding and following cases with NTDs.

Deb et al. [62] observed that mean Hcy levels were elevated in cases (15.71 ± 8.35 μmol/L) of NTD births compared to controls (12.87 ± 5.95 μmol/L) but were lower among the non-vegetarians (13.55 ± 6.64 μmol/L) than the vegetarians (14.78 ± 7.93 μmol/L). Due to low intake of vitamin B12, vegetarianism increased the risk of NTDs 1.6-fold (95% CI = 1.0–2.7), while folic acid supplementation demonstrated a protective effect for conceptions (OR = 0.59; 95% CI = 0.3–0.9). One meta-analysis [51] examinated 17 articles involving 3237 subjects and concluded that mothers with NTD offspring had significantly a higher mean log plasma Hcy level than mothers with normal offspring (log WMD: 0.06; 95% CI: 0.02–0.09, *p* = 0.001), but no positive association was found in studies conducted in countries with folic acid fortification.

Besides environmental and genetic factors, epigenetic mechanisms also play an important role in the etiology of multifactorial diseases such as NTDs. Iacobazzi et al. [63] described that when Hcy occurs at high levels, it competes with SAM for binding sites on DNA methyltransferase, leading to DNA hypomethylation with consequences for epigenetic programming. The reduction of DNA methylation during embryogenesis might be a genetic risk for NTDs. Zhao et al. [64] demonstrated a SAM/SAH reduction in mothers with NTD-affected offspring. Interesting evidence of the role of HHcy in determining neural tube defects (NTDs) recently emerged in a study by Zhang et al. [65] that suggested that higher levels of Hcy contribute to NTDs through upregulation of histone H3 *N*-homocysteinylation at Lys-79, leading to abnormal expression of selected NTC-related genes.

## 3. Children and Adolescents

Bailey et al. [66] defined cut-off levels for Hcy in healthy children and adolescents in the Canadian Laboratory Initiative for Pediatric Reference Intervals (CALIPER) program. The cut-off levels were as follows: 1–7 years: 2.7–7.6 μmol/L; 7–12 years: 3.4–8.4 μmol/L; 12–15 years: 4.7–10.4 μmol/L (males), 4.1–10.4 μmol/L (females); 15–19 years: 5.5–13.4 μmol/L (males), 4.9–11.9 μmol/L (females). As suggested by some authors [38,66,67], Hcy levels in children and adolescents are influenced by age, sex, ethnic differences in genetic polymorphisms of MTHFR, vitamin B12 deficiency, folate metabolism, and the laboratory test used to evaluate said levels. Table 1 collects pediatric reference intervals from several studies, showing the changes of Hcy and giving a picture of the variability found in healthy subjects. The majority of studies conducted on healthy children and adolescents have revealed that Hcy levels increase slightly as a function of age, ranging from 4.6 to 10.2 (μm/L) for 4–5 years and 16–19 years, respectively, in males, and from 3.8 to 8.33 (μm/L) for 9 years and 15-19 years, respectively, for females [68,69,70,71]. No trend with age was observed by Dávila-Rodriguez et al. [72], even though they found higher levels of Hcy in a group of 56 Mexican children (age: 2–9.99)―up to 11.94 ± 2.03 μm/L―compared to those of non–Hispanic Caucasican children reported in other studies and of other Latin-American pediatric groups [73,74]. Many studies have found gender differences in Hcy levels, with slightly higher Hcy levels in boys than in girls. This gender effect is enhanced during and after puberty (>15 years) [68,70,73]. The effect of puberty in increasing Hcy levels may be due to the different creatine turnover rates, with increased creatine production in boys related to their increase of muscle mass. Demand for creatine biosynthesis in men is usually greater than in women, and the link between creatine biosynthesis and Hcy metabolism is the enzyme guanidinoacetate methyltransferase [75]. Furthermore, increased Hcy levels were previously associated with increased androgen levels in boys [76], whereas estrogen levels were negatively correlated with Hcy levels in women [77]. Some authors have underlined that gender differences in Hcy concentrations may also be due to a greater efficacy in the remethylation and transmethylation phases in women, though the mechanism for this is still unknown. These phases represent important steps in Hcy metabolism, and the heightened efficacy of these steps in women may result in differences between males and females [78].

In addition to age and sex, folate and vitamin B status are the most important determinants of Hcy levels in children and adolescents. There are consistent reports of a strong relationship between HCy and serum folate in healthy children and adolescents, as reported by Gonzàlez-Gross et al. [79] (*r* = −0.328); meanwhile, the relationship with vitamin B12 biomarkers is weaker (*r* = −0.219 and −0.201 for cobalamin and holo-transcobalamin, respectively) (all *p* < 0.01) [79]. Other authors have obtained same results [68,69,75,80]. An interesting study [81,82] conducted on 26 hyperhomocysteinemic children (aged 6–15 years), 20 supplemented with 5mg oral of folic acid twice a week for 2 months and 6 children used as controls, showed that supplementation significantly decreased serum Hcy (*p* < 0.001) levels as well as systolic and diastolic pressure (*p* < 0.001 and *p* = 0.045, respectively) [78]. Moreover, significant improvements of total cholesterol levels from 183.8 (115–296 mg/dL) to 160.8 (109–265 mg/dL) (*p* < 0.05) were also observed [82].

The effect of lifestyle factors (e.g., diet and smoking) is poorly documented in children and adolescents. There is a positive association between Hcy level and sugar intake [83], as well as smoking habits [84]. A cross-sectional study on 483 subjects, 7–15 years of age and of both sexes [85], demonstrated that high serum Hcy levels were associated not only with age ≥12 years (prevalence ratio (PR) = 2.56; *p* < 0.01), being male (PR = 3.74; *p* < 0.01), high blood pressure (PR = 1.97; *p* < 0.01), low HDL-c levels (PR = 1.21; *p* = 0.03), high triglyceride levels (PR = 1.62; *p* = 0.03), and being overweight (PR = 2.32; *p* = 0.02), but also with poor intake of foods (PR = 1.46; p = 0.02) that protect against HHcy, such as dark green vegetables, whole and enriched grain products, legumes, and citrus fruit. Arouca et al. [86] found no positive correlation between high levels of Hcy and low adherence to the Mediterranean Diet or other “antioxidant diets” among adolescents in the Helena study.

Hcy concentration should also be expected to be affected by genetic background; however, data are scarce in children. In children (6–11 years) with familial hypercholesterolemia, an association between C677T mutation in the *MTHFR* gene and low serum folate and increased Hcy was found during cholestyramine treatment [87]. In a group of healthy young people (aged 3–18 years), the authors concluded that *MTHFR* genotype played a significant role in determining Hcy concentrations in subjects nutritionally stressed and aged over 10 years old [88].

The effect of *MTHFR* 677C>T polymorphism is enhanced by folate plasma levels. Caldeira-Araújo et al. [71] found that in a group of 9 year old children with high plasma folate levels, Hcy plasma concentrations did not differ significantly among the *MTHFR* 677 genotypes; the 17 year old adolescent subjects bearing the *MTHFR* 677TT genotype (with high folate levels) displayed significantly higher Hcy concentrations than those bearing the wild-type genotype. These results suggest that folate levels can modulate the expected correlations between genotype and Hcy levels.

### 3.1. HHcy and Cardiovascular Diseases

In recent years, research studies have investigated the role of HHcy levels in pediatric cardiovascular diseases, both to determine their relationship with the severity of diseases and to use Hcy as prognostic biomarker for increased risk of cardiovascular diseases. Recently, studies on cardiovascular disease in children and adolescents have examined postural tachycardia syndrome (PoTS) [89], congenital anomalies (congenital adrenal hyperplasia) [90], and heart failure [91]. PoTS is a common functional cardiovascular disease in children and adolescents, representing one of the main causes of syncope in pediatric populations, and it accounts for about 32.2% of all cases [92,93]. PoTS syncope is more common in adolescent females and often occurs after menarche [94]. The symptoms of PoTS include nausea, headache, palpitations, dizziness, chest discomfort, blurred vision, shortness of breath, and, in severe cases, syncope. The etiology of PoTS remains quite unclear; some authors suggest that this syndrome is mainly induced by an autonomic nerve dysfunction, and in particular by an increased baroreflex sensitivity (BRS), and BRS is correlated with PoTS symptoms [95,96]. Some studies on hypertensive patients have shown a relationship between HHcy levels and increased BRS [97], probably due to the regulation of oxidative stress mediated by Hcy. Based on these observations, Li et al. [89] demonstrated in a case–control study the role of Hcy in modulating PoTS symptoms in children, probably by interfering with BRS. The study was conducted on 35 children with PoTS (15 males and 20 females, age 6–13 years with a median of 11 years) and 30 controls (12 males and 18 female, age, 9–12 years with a median of 10 years). The results showed that Hcy plasma levels were significantly higher in the PoTS-affected subjects than the control group (9.78 (7.68, 15.31) μmol/L vs. 7.79 (7.46, 9.63) μmol/L, *p* < 0.05) as well as revealing a positive correlation with the PoTS symptom scores. Hcy can play an important role in modulating PoTS; nevertheless, further studies are needed to clarify how it could be used as an effective index to predict the short-term outcome of PoTS in children. At present, Hcy levels are suspected to play a role in etiology and control of these results in some cardiovascular diseases like PoTS, and some studies have shown that measurement of Hcy in children with acute heart failure (HF) and congenital adrenal hyperplasia (CHA) can be a very useful biomarker to predict the course of these diseases [90,91]. High levels of Hcy seem to determine adverse cardiac remodeling, both directly affecting myocardium or as a result of an independent vascular effect [98]. In a case–control study, El-Amousy et al. [91], investigating the prognostic value of Hcy levels and of highly sensitive troponin T, found children with HF with significantly higher Hcy plasma levels before HF treatment (11.150 ± 1.960 μmol/L), compared with after treatment (9.030 ± 1.616 μL·mol/L) and with the control group (6.69 ± 0.97 μL·mol/L). In patients with congestive heart failure, the authors suggested cut-off points of Hcy > 8.1 µmol/L and Hs-cTnT > 10 pg/mL levels as diagnostic predictor. The authors reported both plasma Hcy and serum hs-cTnT levels to be significantly (*p* < 0.001) increased in adverse outcomes (Hcy 13.23 ± 1.03 µmol/L; Hs-cTnT 110.50 ± 10.98 pg/mL) compared to favorable outcomes (10.96 ± 0.87 µmol/L and 69.78 ± 16.35 pg/mL, respectively). In addition, a significant increase of the mean levels of hs-cTnT and Hcy was associated with severity of HF (Class IV patients vs. Class III and Class II patients (*p* = 0.001). Results showed Hcy levels were correlated with clinical and echocardiographic data recording severity of HF, confirming previous findings on HF [99,100]. In this study, plasma Hcy and serum hs-cTnT levels were highlighted as good prognostic biomarkers in children with congenital heart diseases.

A case–control study conducted on 36 children with congenital adrenal hyperplasia (CAH), and 36 controls [90] evaluated serum levels of Hcy in relation to relation to carotid artery intima-media thickness (CA-IMT) and left ventricular (LV) function. Compared to the controls, high serum Hcy levels and positive correlation with CA-IMT and left ventricular (LV) function were present in CAH-affected children. Hcy levels were also significantly correlated with systolic (OR = 2.2; 95% CI: 1.10–1.18; *p* = 0.01) and diastolic blood pressures (OR = 2.9; 95% CI: 1.45–2.4; *p* = 0.01) and atherogenic index (OR = 2.6; 95% CI: 1.33–2.89; *p* = 0.01). The authors concluded that serum Hcy levels in children with CAH should be used as a biomarker for an increased risk in developing LV dysfunction and subclinical atherosclerosis.

As observed for autism spectrum disorder (ASD) studies, these studies had some limitations including relatively small number of patients and lack of evaluation of folate and B12 levels, or any possible mutation for genes encoding enzymes involved in the Hcy pathways. All these factors might lead to some bias and preclude an exhaustive evaluation of the role of Hcy in these diseases; thus, further studies are needed to overcome these shortcomings.

However, with regards to Hcy and cardiovascular disease in children and adolescents, some interesting conclusions emerged from the systematic review of Leal et al. [101], conducted on studies published between 1997 and 2011 on this topic. The authors underlined these fundamental conclusions: (a) HHcy is positively correlated with *MTHFR* 677T and heterozygous *MTHFR* 677T/1298C, with cardiovascular disease, and with low levels of folate and B12; (b) Hcy levels increase with age, being higher in adolescents than in children, and are higher in males than in females; (c) a possible association exisits between high Hcy levels in children and adolescents and a positive parental history for cardiovascular diseases; (d) there is a positive correlation between high Hcy and overweight and obesity in children and adolescents, with an increased risk for cardiovascular diseases (CVD). The correlation between HHcy levels and increased risk of CVD in children and adolescents is not yet well clarified. In their review, Leal et al. [101] found many studies concluding a positive parental history of CVD and HHcy in children and adolescents and studies showing HHcy levels in overweight or obese subjects [102]. The authors [101] suggested the need to investigate HHcy in conjunction with other parameters such as insulin levels (which have an inverse relationship with Hcy) and renal function to clarify the pathophysiological mechanism by which HHcy can increase the risk of CVD.

### 3.2. Obesity

In regards to the relationship between overweight and obesity and HHcy in children, a recent study [103] examined 138 children (46 normal weight, 40 overweight, and 52 obese) and observed that the median Hcy levels in overweight (16.7μmol/L, range: 11.2–2.5) and obese children (16.6 7 μmol/L, range: 13.3–22.4) were significantly higher than in normal weight children (7.3 μmol/L, range 5.5–10.5), *p* = 0.001. Debohkordi et al. [104], in a previous randomized double-blind controlled clinical trial on 60 obese and overweight children aged 5–12 years demonstrated that supplementation of 1 mg folic acid per day or 5 mg folic acid per day for 8 weeks had a positive effect on the reduction of HHcy and insulin resistance. Folic acid supplementation might represent a good strategy for decreasing high Hcy levels and related cardiovascular risk in obese and overweight children. However, further studies should be carried out with suggested lower levels of supplementation. In fact, the levels used in the randomized trial of Debohkordi et al. [104] were in excess of the upper levels of folic acid suggested by Dietary Reference Intakes [105,106] and are not safe for long-term supplementation. Overproduction of Hcy could be harmful and is a possible cofactor in the etiology of multifactorial diseases or a biomarker of chronic and acute inflammatory status and diseases, increasing the risk of other diseases and complication. The majority of studies on the relationship between Hcy levels and diseases in children and adolescents to date have been observational or case–control studies with a small number of subjects. At the present time, they provide data supporting a further in-depth investigation of the biochemical and metabolic mechanisms between metabolic Hcy and the etiology of some diseases.

### 3.3. HHcy and Renal Diseases

Hcys levels are dependent in children and adolescents with renal diseases. The kidney is a major organ that metabolizes Hcy, and most patients with chronic renal diseases show HHcy. HHcy in chronic renal disease is due to a decrease in intrarenal HCy clearance and/or a decrease in extrarenal Hcy clearance [107]. HHCy in renal diseases shows a similar pattern to atherosclerosis, i.e., production of lipid-laden macrophage, presence of cholesterol and cholesterol ester, intima thickening, elastic lamina disruption, and luminal platelet accumulation [108,109]. Nephrotic syndrome (NS), a common pediatric kidney disease, is associated with HHCy due to inhibition of the homocysteine remethylation process or disruption in cysteine clearance [110]. Most NS patients are resistant to steroid treatments and have evolved into focal segmental glomerulosclerosis (FSGS) due to the pathogenic effect of HHcy, which causes podocyte injury and glomerulosclerosis [111,112].

Merouani et al. [113] examined 29 children affected by different degrees of chronic renal failure (15 dialyzed, 14 not dialyzed) compared with 57 age-matched healthy children. Hcy concentrations were significantly higher (*p* < 0.0001) in patients than controls (17.3 and 6.8 μmol/L respectively) and 62.0% of patients and 5.2% of controls had HHcy (>95th percentile for controls: 14.0 μmol/L). Inversely, folate levels were significantly lower (*p* < 0.01) in children affected by chronic renal failure (9.9 nmol/L) than in controls (13.5 nmol/L). The authors [110] found a higher prevalence of HHcy in dialyzed children than in nondialyzed patients (87% vs. 35%), and higher levels of HHcy in dialyzed children with *MTHFR* mutations (28.5 μmol/L) compared to nondialyzed patients with the mutation (10.7 μmol/L *p* < 0.002). Child patients (*n* = 12, aged 11.0 ± 3.5) with acute glomerulonephritis had higher Hcy levels (from 9.4 ± 3.3 to 13.5 ± 2.8 μmol/L) compared to controls (*n* = 15, 8.4 ± 2.4 μmol/L) and lower vitamin B6 and RBC folate levels (*p* < 0.01) [104]. HHcy was also found in 26 renal transplanted children and adolescents (12.9 ± 4.8 μmol/L), with 73% of these patients showing Hcy levels above the 97th percentile of healthy children. Plasma HCy in these subjects was negatively correlated with plasma vitamin B12 (*r* = 0.40, *p* < 0.05) and creatine clearance (*r* = 0.53 *p* < 0.005), and plasma Hcy levels were higher in patients with a *MTHFR* 677TT/1298AA genotype [114,115]. The above-cited studies did not investigate the nutritional intake of folate, other B vitamins (B6 and B12), or of other micronutrients in children with renal diseases. As suggested by Viroonudomphol et al. [107], an adequate intake of folate, B6, and B12 with an increase of fruit, vegetables, wholegrain cereals, and legumes and/or an appropriate targeted supplementation could increase vitamin B status, reducing HHcy risk of end-stage renal diseases and reducing clinical complications and risk of cardiovascular events in renal transplant patients.

### 3.4. HHcy and Autism Spectrum Disorder (ASD)

Autism spectrum disorder (ASD) is a neurodevelopmental syndrome with a very complex pathogenesis, not yet completely understood even though gene–gene and gene–environment interactions that might play roles have been suggested [116]. Several studies have observed folate, Hcy, and glutathione metabolism anomalies in many children affected by ASD [117,118,119]. A great deal of research has demonstrated high levels of Hcy and low levels of folate, vitamin B6, and vitamin B12 in children affected by ASD [116,117,118,119,120,121]. In a case–control study, Altun et al. [121] focused on 105 children (3–12 years; 60 with confirmed ASD and 45 healthy controls) and observed significantly increased levels of Hcy in affected children compared to the control group (8.90 ± 0.19 nmol/L and 7.46 ± 0.21 nmol/L, respectively; *p* < 0.001). In contrast, the levels of vitamin B6, folate, and vitamin B12 were found to be significantly lower in the patients with ASD (25.17 ± 3.64 ng/mL, 121.16 ± 8.04 pg/mL, and 181.5 ± 41.61 pg/mL, respectively; 53.06 ± 7.95 ng/mL, 172.31 ± 17.19, pg/ml, and 382.06 ± 71.34 pg/mL, respectively; *p* < 0.001). Other studies confirmed higher levels of Hcy in ASD patients compared to healthy children in serum, plasma, and urine [118,122,123,124] with respect to control groups. In a clinical update, Fuentes-Albero et al. [120] reviewed current evidence aligning the severity of autism symptoms and HHcy. As for neurodegenerative diseases, one hypothesis of the significance of HHcy in the etiology of ASD is that Hcy is a powerful excitotoxin, and its metabolic products can cause damage in some proteins, generating “abnormal toxin proteins” and inducing an autoimmune response [125]. Some studies on the neurotoxicity of Hcy have demonstrated that Hcy can induce neuronal damage, causing activation of *N*-methyl-d-aspartate (NMDA) receptors and cell loss through excitotoxicity as well as apoptosis [124,125,126,127,128,129,130,131]. Additionally, Fulceri et al. [127] observed that autistic children often have gastrointestinal disorders and/or are picky eaters, and these factors could lead to an inadequate dietary intake, altered nutrient absorption, and micronutrient deficiency, particularly vitamin B deficiency, which causes increased plasma Hcy levels and worsening of autism. Along with HHcy, methylation impairment and decrease in glutathione redox status may be cofactors in autism pathology [128,129,130].

On the basis of the relationship between low folate levels, HHcy, and autism, Sun et al. [131] in an intervention study demonstrated that supplementation with 800 μg/day of folic acid in children (mean age 52.00 ± 13.13 months) improved autism symptoms related to receptive language, cognitive verbal/preverbal, sociability, affective expression, and communication, and that this treatment also improved the concentration of folic acid, reduced Hcy plasma levels, and normalized glutathione redox metabolism. These results, according to Józefczuck et al. [132], suggest serum Hcy levels as a useful biomarker for diagnosis of autism spectrum disorder in children as well as for the efficacy of treatment.

### 3.5. HHcy and Attention Deficit Hyperactivity Disorder (ADHD)

Interesting observations regarding levels of Hcy in relation to attention deficit hyperactivity disorder (ADHD) emerged from the study of Altun et al. [133] showing an opposite behaviour of Hcy levels with respect to ASD. ADHD is a neurobehavioral disorder characterized by symptoms of inattention and impulsivity, and is widespread among children. As ASD, ADHD is a multifactorial disorder and its etiology has not been clarified. Some studies have investigated the role of folate, B12, B6, and Hcy levels in many psychiatric diseases and depression [134,135,136,137] for the fundamental roles of these compounds in the carbon transfer metabolism, the necessary pathway to produce serotonin, catecholamines, and other monoamine neurotransmitters. These studies correlated high levels of Hcy and low levels of folate, B12, and B6 with the etiology and/or symptoms of diseases. Interesting results emerged from a study [133] conducted on 30 children and adolescents aged 6–15 years with ADHD and 30 controls (also aged 6–15). This study demonstrated lower folate levels in ADHD affected children with respect to the controls, as already observed in adults affected by ADHD [138], with a different behaviour of Hcy plasma levels observed in the majority of the studies. The authors explained that low Hcy levels could be an index of an excessive transformation to cystatin in the trans-sulfuration pathways for glutathione, taurine, and sulfate production, as already observed in Down’s syndrome patients [139]. Certain conditions or diseases that increase oxidative stress in the human body may increase hepatic glutathione synthesis, determining a decrease of plasma Hcy levels, which are critical for potential total body glutathione status. Low plasma levels of Hcy determine a reduction in cysteine and a very limited capacity of individuals to respond to oxidative stress [133]. With respect to the results of many studies indicating the correlation of low levels of vitamin B6, folate, and vitamin B12 with higher Hcy plasma levels, Altun et al. [133] observed lower levels of these compounds in ADHD patients compared to the control group. Thus, further studies are required to clarify the specific relationships between Hcy, folate, vitamin B12, and pyridoxine vitamin B6 levels for the assessment of ADHD.

## 4. Elderly 

HHcy is common among elderly people and is often associated with cardiovascular diseases (CVD) (including coronary artery disease, stroke, and peripheral vascular disease), cognitive impairment, dementia, depression [140,141,142], and osteoporotic fractures [143]. In a large cohort of community-dwelling older men, HHcy appeared to be predictive of all-cause mortality, independent of frailty [136], an age-related clinical state characterized by a global impairment of physiological functions and involving multiple organ systems [144]. A meta-analysis of prospective studies indicated that an increase of 5 μmol/L Hcy increased by 27% the risk of all–cause mortality, by 32% the risk of cardiovascular mortality, and by 52% coronary heart disease mortality, and the risk was more elevated in elderly population [145]. In a long-term study [146] on middle-aged and elderly subjects, increased levels of Hcy (≥10.8 μmol/L) were significantly associated with all-cause mortality (*p* < 0.001) and CVD (*p* < 0.001) compared with decreased levels (Hcy < 10.8 μmol/L). As already described above, the plasma concentration of Hcy is the result of a close relationship between dietary habits, lifestyle, renal function, and predisposing genetic factors. The latter represent only a small fraction of the causes of HHcy. High levels of Hcy are frequently associated with a low dietary intake of folate and other B vitamins (B2, B6, and B12), and several studies have identified links between HHcy, some pathologies, and deficiencies in these micronutrients [147,148]. As an example, a study performed on 1350 elderly Taiwanese subjects aged 65–90 years with normal renal function showed that low plasma levels of folate and vitamin B6 or vitamin B12 were associated with a 3–5-fold increased risk of HHcy (>15 μmol/L) compared to patients with only one B vitamin insufficiency [147]. On the other hand, HHcy often occurs independently of vitamin status [149,150,151]. A recent review [152] reporting the impact of HHcy on aging and related diseases emphasized the importance of screening elderly subjects over 60 years for raised Hcy levels and of evaluating folate or vitamin B12 status for accurate nutritional requirements. The authors also underlined that supplementation with B vitamins, vitamin D, and antioxidants such as via a vitamin-rich diet “must be recommended to all people over 65 years of age”.

### 4.1. Cardiovascular Diseases

Some case–control studies have shown that 10% of all coronary artery disease is attributable to HHCy, and that an increase of HCy of 5 µmol/l raises the risk of ischemic heart disease by 84% [153]. Tang et al. [154] correlated plasma Hcy levels in patients (aged > 60 years old) with various ischemic cerebrovascular diseases (transient ischemic attack (TIA), large-artery atherosclerosis (LAA), and small-artery occlusion (SAO)). HHcy was closely related to occurrence of TIA; in fact, Hcy levels in the TIA group (17.05 ± 5.36 μmol/L) were significantly higher than those of the LAA (14.39 ± 5.22 μmol/L, *p* = 0.002) and SAO (13.54 ± 3.72 μmol/L, *p* = 0.000) groups, while no significant difference was found between the LAA and SAO groups. HHcy could play a key role in the occurrence and development of TIA through free radical production by Hcy oxidation. The high toxicity of these free radicals to vascular endothelial cells, could promote an increased synthesis of oxidized low-density lipoprotein (OxLDL). Thus, atherosclerosis progression could be promoted by induced inflammation and plaque instability. Cao et al. [155] sustained the involvement of HHcy in the pathological manifestation of atherosclerosis. In their study, the Hcy levels (ranged 5–37.1 μmol/L) of 1357 patients (aged 31–90 years) were significantly higher in the carotid/intracranial artery stenosis group than in the control (non-stenosis) group. The involvement of HHcy was probably via worsening the inflammation mechanism that led to the increase of unstable carotid plaques and the occurrence and development of acute cerebral infarction. Since inflammatory response plays an important role in the process of ischemic stroke, Hcy increases the risk of ischemic stroke. Recently, the results observed in a metabolomic study [156] conducted on 65 male patients with severe carotid artery stenosis (aged 60.2 ± 5.9 years) highlighted significantly higher levels of plasma Hcy but lower levels of cholesterol, high-density lipoprotein, and hemoglobin than 65 male control subjects (aged 63.3 ± 5.2) (*p* < 0.05).

As already mentioned, HHcy can be reduced by vitamin B group supplementation. Thus, if the relationship between Hcy and CVD is causal, supplementation could be expected to reduce the risk of CVD and stroke and this could be tested in randomized controlled trials. Some meta-analyses have been conducted to evaluate the effects of folic acid supplementation on stroke risk [157,158]. Lee et al. [157] identified 13 randomized controlled trials with a total of 39,005 adult participants and observed that folic acid supplementation did not reach statistical significance in reducing the risk of stroke (RR = 0.93; 95% CI, 0.85–1.03; *p* = 0.16). When folic acid supplementation was associated with B6 and B12, a potential mild benefit was observed (RR = 0.83; 95% CI, 0.71–0.97; *p* = 0.02), especially in men (trials with ratio men:women >2; RR = 0.84; 95% CI, 0.74–0.94; *p* = 0.003). Zeng et al. [158], examining 14 randomized trials with a total of 39,420 adult patients, stratifying trials according to the folate fortification, concluded that folic acid supplementation “might have a modest benefit on stroke prevention in regions without folate fortification”. In a study conducted on 390 healthy Chinese subjects aged 60–74 years, supplemented with vitamin C alone (control group) or vitamin C and B vitamins (treatment group) for 12 months, the authors reported that the Framingham risk score, a predictor of cardiovascular disease risk, was particularly reduced in elderly subjects with folate deficiency [159]. As pointed out by Ostrakhovitch and Tabibzadeh [152], although no consensus has yet been reached regarding the effectiveness of HHcy correction in preventing the development of vascular disease, Hcy-lowering therapy “might useful in lowering the risk of inflammation and vascular risk factors”. Several hypotheses have been advanced in the literature. Folate and vitamin B nutritional status could influence the response to supplementation. Loscalzo et al. [160] hypothesized that high levels of unmetabolized folate might also interfere with the effectiveness of B vitamin supplementation. Zhao et al. [161] underscored the importance of individual genetic background and nutritional status of patients. Their analysis of the data of 20,424 hypertensive adults enrolled in the China Stroke Primary Prevention Trial observed that in Chinese hypertensive patients, the effect of Hcy on the first stroke was significantly modified by the methylenetetrahydrofolate reductase C677T genotype and folic acid supplementation. In fact, folic acid intervention significantly reduced stroke risk in participants with the *MTHFR* CC/CT genotype and high Hcy levels (tertile 3; hazard ratio, 0.73; 0.55–0.97). It is also interesting to note that folic acid supplementation was significantly more effective in individuals with the *MTHFR* TT genotype without HHcy (<12.8 μmol/L), compared with individuals with HHcy in the same genotype. The authors speculated that differences in folate status in MTHFR enzyme activity such as long-standing injury between participants with CC/CT genotype and those with the TT genotype may have been responsible for the observed results, and speculated that higher doses of folic acid or longer supplementation periods would be needed.

### 4.2. Neurodegenerative Diseases

In addition to the known influence of Hcy on CVD, numerous diseases of the nervous system, such as cognitive decline, Alzheimer’s disease (AD), and Parkinson’s disease (PD) are correlated with HHcy. The risk of Alzheimer’s disease doubled with a plasma homocysteine level greater than 14 µmol/L, as observed by Seshadri et al. [162] in an 8 year follow-up study on 1092 subjects without dementia (667 women and 425 men; mean age, 76 years). Numerous mechanisms by which Hcy may act to promote these conditions have been proposed, such as excitotoxicity for neurons, increased generation of reactive oxygen species (ROS) production, inflammatory processes, and long-term levodopa treatment [35,48]. Hooshmand et al. [163], examining 2570 elderly individuals (mean age, 73.1 ± 10.4 years) over a 6 year period found that the methionine to Hcy status was associated with dementia development and structural brain changes, suggesting that a higher methionine to Hcy ratio may be significant in decreasing the rate of brain atrophy and decreasing the risk of dementia in older adults. Differences in vitamin B status can influence the grade of disease, although contrasting results are present in the literature. In the VITAGOG study [164], a two year supplementation with B vitamins was associated with a 30% reduction in Hcy and an improvement of cognitive decline. Douaud et al. [165], in a study performed on elderly subjects with increased dementia risk, found that B-vitamin supplementation slowed the atrophy of specific brain regions that are key components of Alzheimer’s disease and that are associated with cognitive decline. Similarly, other studies [166,167] have found that supplements containing folate, vitamin B6, and vitamin B12 in patients with HHcy could reduce Hcy levels and improve their cognitive function. Further studies have confirmed the efficacy of some vitamins in lowering Hcy, but not in the attenuation of pathological symptoms. In a meta-analysis of selected randomized controlled trials, including in total 679 patients with cognitive decline secondary to Alzheimer disease or dementia (age range: 74.6–79.1 years), upon intervention with folic acid along with vitamin B12 and/or B6, the authors observed a significant reduction in Hcy levels in the treated patients compared to the controls (pooled difference in means −3.626 µmol/L), but it was not associated with cognitive improvement [168]. In healthy men and women aged 55–65 years, after a multivitamin supplementation for 16 weeks, an improvement of blood biomarkers (reduced Hcy, increase of vitamin B6 and vitamin B12 levels) without effect on cognitive decline was observed [169]. Ma et al. [148] investigated the relationship between cognitive impairment and Hcy, folate, and vitamin B12 in a case–control study on elderly Chinese subjects (age ≥ 65 years: 112 with mild cognitive impairment (MCI), 89 Alzheimer’s disease (AD) patients, and 115 healthy control). Serum folate and vitamin B12 levels were significantly lower and the plasma Hcy level was higher in patients than in healthy controls, but no association existed between low vitamin B12 levels and AD or MCI (*p* > 0.05). The authors suggested that further studies using holotranscobalamin as a marker of vitamin B12 deficiency will be performed. Harrington [170] pointed out that “no single laboratory marker is suitable for the assessment of B12 status”, and holotranscobalamin can be a more reliable marker of B12 status than total vitamin B12 alone. The same author suggested using “algorithms or the combination of multiple markers to mitigate inherent limitations of each marker when used independently”. In a longitudinal cohort study on 38 subjects (men aged 81.3 ± 6 years) affected by AD and followed for an average period of 13 months, significant correlations between Hcy and plasma folate (*r*_s_ = −0.58, *p* < 0.001) and vitamin B12 (*r*_s_ = −0.42, *p* < 0.01), and between Hcy and dietary intake of folate (r_s_ = −0.34, *p* < 0.04) were found at baseline. A decline of cognitive status associated with an increase of plasma level of Hcy (p=0.006) was observed at follow-up, but no association with vitamin B status was found [149]. Similarly, Bonetti et al. [150], in 318 elderly subjects (age ≥ 65 years) (44 normal cognition, 127 with cognitive impairment, 147 with dementia) found that HHcy (>15 μmol/L) was associated with a higher prevalence of cognitive and functional impairment and dementia (odds ratio (OR) = 1.98, 95% confidence interval (CI) = 1.13–3.48) independent of B group vitamin status. The authors suggested that HHcy cannot be explained only by vitamin status and that other dietary and non-dietary factors may contribute [150,151]. Lifestyle, coffee consumption, smoking, and alcohol use seemsto be associated with HHcy in Parkinson’s disease patients treated with levodopa [171]. Oulhaj et al. [172], analyzing the data of the VITACOG study, found that different omega-3 fatty acid status results in the trial participants influenced the effect of vitamin B treatment, and suggested that the interaction between the two nutrients on cognition and brain atrophy must be considered and that a complementary supplementation could delay the conversion from mild cognitive impairment to dementia.

### 4.3. Osteoporosis

Low bone mineral density (BMD) is a common geriatric problem, and it has been hypothesized that the metabolism of HCy is involved in osteoporosis, increasing the risk of osteoporotic fracture [1]. In the Framingham study [173] on 825 men and 1174 women aged from 59 to 91 years, with a follow up 12.3 years for men and 15.0 years for women, it was observed that HHcy (>20 μmol/L for men and >18 μmol/L for women) is associated with higher risk of bone fracture, almost 4 times as high for men and 1.9 times as high for women. Similarly, van Meurs [174] showed in two independent prospective studies that in 2406 subjects aged 55 years and older, the overall multivariable-adjusted relative risk of fracture was 1.4 (95% confidence interval, 1.2 to 1.6) for each increase of 1 S.D. in the natural-log-transformed Hcy level. An Hcy level in the highest age-specific quartile was associated with an increase in the risk of fracture twice as high as the risk in the lower three quartiles. Several mechanisms by which Hcy may act to promote the pathological status have been proposed [175,176,177,178,179,180]. In particular, as reviewed by Toohey et al. [181], HHcy can cause connective tissue pathology by conversion of Hcy thiolactone into mercaptopropionaldehyde via several routes, such as ascorbic acid depletion influencing collagen synthesis, interfering directly with collagen synthesis, causing abnormal cross-linking of collagen molecules, and triggering autoimmune response. Moreover, oxidative damage and the adverse effects of Hcy on endothelial-derived nitric oxide could increase risk of fracture [182,183,184]. *MTHFR* polymorphism C667 T could also be involved in the increase of osteoporotic fracture incidence [181], and a recently published meta-analysis showed an association between MTHFR C667 T polymorphism and reduced bone mineral density in the lumbar spine and femoral neck in Caucasians, post-menopausal women, and men, and with total BMD in women [185]. Ahn et al. [186] reported an association in 301 postmenopausal women between osteoporotic vertebral compression fractures and *MTHFR* and *TS* gene polymorphisms in the 3′-UTR region. Concerning the effect of supplementation, a systematic review with meta-analysis [187] concluded that vitamin B12 status may be associated with fracture risk, and evidence for an association between folate status and fracture risk is scarce. In the B-PROOF study [188,189], a randomized placebo-controlled trial of a 2 year intervention of vitamin B12 and folic acid supplementation in 2919 elderly subjects (aged ≥65 years) with moderate HHcy produced no observable effect of treatment on bone mineral density despite a significant reduction of Hcy (−4.2 ± 3.0 μmol/L, *p* < 0.001). Similarly, other studies have observed no association of Hcy-lowering treatment with B vitamins, bone turnover, and risk of fracture [187,188,189,190,191,192].

### 4.4. Physical Function

Several studies have highlighted the relationship between Hcy and muscle strength and physical function in the elderly, stressing that the decrease in muscle strength could also play a role in the risk of falls. In the MacArthur Studies of Successful Aging on 499 healthy and highly functional community-dwelling people (aged 70–79 years) [193], the authors observed that subjects with elevated plasma Hcy had an increased risk of functional decline over 3 years, with a 50% higher risk of being in the worst quartile of physical function (odds ratio = 1.5; 95% CI, 1.2–1.9) after multivariable adjustment. Similarly, in the Baltimore Longitudinal Study of Aging [194] on 1101 adults aged 50 years or older, in a follow-up of 4.7 years a significant inverse relationship between Hcy and grip strength was observed in women (β= −005, *p* = 0.031), while in men no significant results were observed, although an increase of 1 μmol/L in Hcy was associated with −0.10 kg decrease in grip strength. Oxidative damage of endothelial cells and deoxyribonucleic acid, telomere loss such as leukoaraiosis induced by HHCy, and bioavailability of nitric oxide [195] could be possible mechanisms involved in the relationship between Hcy and lowered physical function [193]. Physical performance is also related to poor vitamin B group nutritional status; moreover, Ao et al. [196] in an observational study performed on 65 women (aged 84 ± 7.1 years) observed low muscle and hand grip strength in subjects with low serum folate concentration and pointed to the importance of adequate folate nutrition to avoid impaired physical function. However, supplementation studies with these vitamins have reported no effect on physical performance, hand grip strength, and risk of falling, despite a significant reduction of Hcy [188,189]. Vidoni et al. [197], studying 774 adults aged 50 years or older, found that only Hcy and not vitamin B12 was associated with gait speed decline. The authors observed that this loss of association could have been due to the good state of nutrition in vitamin B12 of the studied sample, concluding that more studies are necessary in populations with greater variation in vitamin B12 concentrations, as well as a more in-depth study of the mechanisms. Additionally, Veeeranki [195] in his review on the potential implication of HHcy on muscle function and integrity highlighted the need for studies to provide deeper insights in this area.

## 5. Conclusions

Regarding the different pathological processes that can affect the human body at any stage of life, the main role of Hcy as an involved and predicting factor is clear. Despite extensive studies of Hcy’s role in several pathologies at different life stages, the exact mechanisms of action as well as the homeostasic regulation of this biomarker remain unresolved. Various common research approaches (cohort studies, case–control, randomized) and study designs do not take into account the relationships between nutrient intakes and the molecular and cellular responses of the organism that eventually result in health and disease; the requirements according to specific genetic profiles and the influences of age, sex, activity level, and metabolic status as well plasma nutrient levels might not reflect tissue storage and needs, or more generally the “optimal” biochemical status. Hcy levels during pregnancy even seem to be linked to the geographical, cultural, and social characteristics of populations, while for children and adolescents, regulated diet and modulated lifestyle decrease Hcy levels, and thus the determination of hyperhomocysteinemia in childhood and in adolescence could be an opportunity for a nutritional and lifestyle intervention for primary prevention of diseases. However, the results from clinical trials of Hcy-lowering vitamin treatments have not reached consensus about the effectiveness of vitamin supplementation on the reduction in Hcy levels and the improvement of pathological conditions, especially in elderly patients with overt pathologies. It is possible to speculate that these weaknesses explain why micronutrient supplementation studies continue to yield contradictory findings. The importance of novel experimental designs focusing on intra-individual variability as a complement to the typical case–control experimental designs and the study of interactions between different factors should be emphasized.

## Figures and Tables

**Figure 1 ijms-21-01421-f001:**
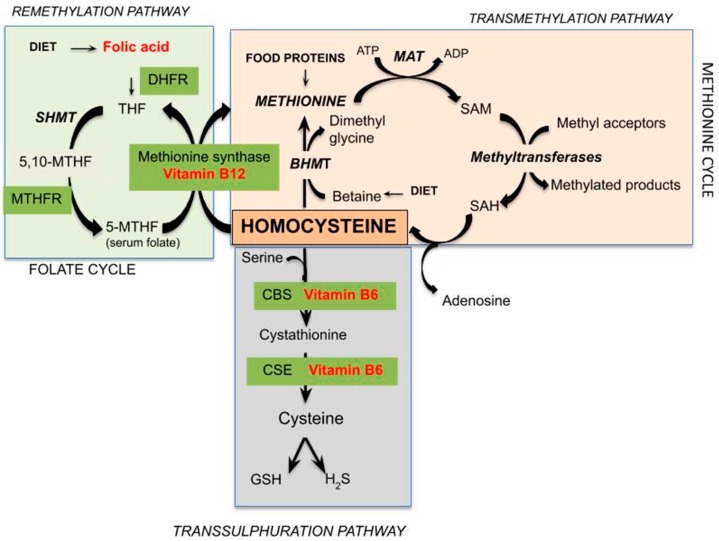
Schematic representation of pathways of Hcy metabolism (DHFR = dihydrofolate reductase; THF = tetrahydrofolate; SHMT = serinehydroxymethyltransferase; MTHF = methylenetetrahydrofolate; MTHFR = 5,10-methylene-THF reductase; ATP = adenosine triphosphate; MAT = methionine adenosyltransferase; ADP = adenosine diphosphate; SAM = *S*-adenosylmethionine; SAH = *S*-adenosylHcy; BHMT = betaine-Hcy *S*-methyltransferase; CBS = cystathionine β-synthase; CSE = cystathionase; GSH = glutathione; H_2_S = hydrogen sulphide).

**Figure 2 ijms-21-01421-f002:**
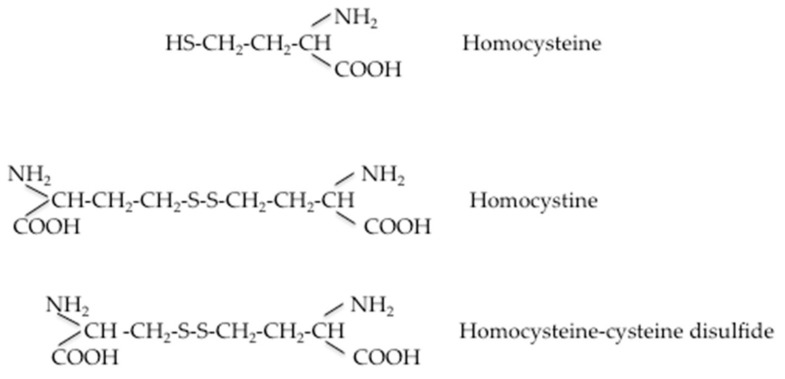
Forms of Hcy present in human plasma.

**Table 1 ijms-21-01421-t001:** Total Hcy levels in healthy children and adolescents in relation to age and gender.

Age Classes (Years)	Male Hcy (μm/L)		Female Hcy (μm/L)		Reference
	n°	Mean ± S.D. (or Range)	n°	Mean ± S.D. (or Range)	
5–9 10–14 15–19	91 115 87	6.30 ^§^ (5.18–7.66) 7.12 ^§^ (5.58–9.01) 9.78 ^§^ (6.70–14.30)	87 114 148	6.11 ^§^ (5.11–7.30) 7.07 ^§^ (5.81–8.60) 8.33 ^§^ (6.29–11.02)	De Laet et al., 1999 [68]
4.00–6.99 7.00–10.99 11.00–14.99 15–18.99	50 128 157 140	5.16 ^§^ (2.7–9.5) 5.59 ^§^ (3.1–9.5) 6.18 ^§^ (2.9–11.5) 8.54 ^§^ (4.1–20.1)	62 108 139 142	4.79 ^§^ (2.2–6.1) 5.69 ^§^ (2.7–10.6) 6.40 ^§^ (3.5–11.8) 7.80 ^§^ (3.9–14.3)	Bates et al., 2002 [69]
4–5 6–11 12–15 16–19	139 161 347 295	4.6 ± 1.1 ^§^ 5.2 ± 1.2 ^§^ 7.2 ± 3.1 ^§^ 8.7 ± 2.8 ^§^	151 174 415 345	4.5 ± 0.9 ^§^ 5.3 ±1.1 ^§^ 6.5 ± 2.6 ^§^ 7.2 ± 2.7 ^§^	Must et al., 2003 [70]
9 17	107 52	3.8 ± 1.8 * 10.2 ± 4.0 *	88 67	3.8 ±1.8 * 7.5 ± 2.2 *	Caldeira-Araújo et al., 2019 [71]
2–3.99 4–6.99 7–9.99	12 18 8	9.78 ± 1.92 ^§^ 11.02 ± 1.60 ^§^ 7.46 ± 1.60 ^§^	6 5 5	11.94 ± 2.03 ^§^ 11.13 ± 1.42 ^§^ 11.13 ± 1.42 ^§^	Dávila-Rodriguez et al. 2010 [72]
M *plus* F <10 10–12 >12	483	7.4 ± 1.82 * 7.5 ± 1.44 * 7.7 ± 1.88 *	483	6.6 ± 1.43 * 7.0 ± 1.76 * 7.1 ± 1.88 *	Ribas De Farias et al., 2017 [73]
14	1815	5.48 ± 1.90 ^§^	1506	5.09 ± 1.78 ^§^	Osganian et al., 1999 [74]
6–17	120	5.7 ± 1.7 *	137	5.5 ±1.6 *	Rauh et al., 2001 [75]
12.5–13.99 14.0–14.99 15.0–15.99 16.0–17.49	139 141 142 130	6.2 ± 2.5 * 6.5 ± 2.6 * 7.2 ± 2.2 * 7.6 ± 2.8 *	139 141 142 130	6.2 ± 2.5 * 6.5 ± 2.6 * 7.2 ± 2.2 * 7.6 ± 2.8 *	Gonzàlez-Gross et al., 2012 [79]

^§^ geometric means ± S.D; * arithmetic means ± S.D.

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
