# Peer review of "Homocysteine: Its Possible Emerging Role in At-Risk Population Groups"

_ijms, 2020, doi:10.3390/ijms21041421_

Round 1

Reviewer 1 Report

This mini review is dedicated to the role of homocysteine (Hcy) in various pathologies occurring at an pregnancy, early and old age. Compared with the previous version (449109), this manuscript has been significantly revised. Revealing the problem of hyperhomocysteinemia in the above groups of people with a wide coverage of the pathology is a rather actual. But, unfortunately, the material presented is not systematized enough, completeness, and depth of research analysis. The review has quite a bit of novelty. In addition to highlighting the well-known problems related to the clinical efficacy of reducing homocysteine ​​levels, it would be desirable to present more detailed considerations of the authors on how to solve these problems. I believe that a major revision of this manuscript is necessary.

Comments
1. From the abstract it is not clear what role homocysteine is in what pathologies is considered. The summary, in fact, does not contain new, important and interesting points and positive conclusions. It is recommended to make a stronger emphasis on the choice of research objects and the breadth of the studied pathologies.
2. Line 15: what does the criterion for the effectiveness of vitamin intake in this context relate to: a decrease in homocysteine levels or clinical indicators?
3. Figure 1: H2S is a by-product of the activity of CBS and CSE and depicting its formation from cysteine as shown in this figure is incorrect.
It is also better to designate enzymes, in contrast to metabolites, with a different font or frame for clarity.
4. line 114: Excess Hcy can also be exported to the extracellular medium via transmembrane transport systems (ASC, L-, XAG transporters, see for example [Jiang X., // Arterioscler. Thromb. Vasc. Biol. - 2007 - v. 27 (9) - p. 1976-1983].
5. line 118: “ Furthermore, the authors [30] hypothesized that homocystine and mixed disulfides derived from plasmatic Hcy oxidation can enter kidney and liver cells via another yet unidentified transporter, where the reducing environment of cells can regenerate Hcy.”
Also, part of homocysteine in a protein-bound form, mainly albumin, enters the cells by endocytosis of these proteins.
6. line 209: “Children with CHD had higher plasma SAM (131 vs. 100 nmol/L) and N,N‐dimethylglycine betaine (DMG)”
DMG or betaine?
7. line 205-216: does not explain how these studies are related to the HHcy problem during pregnancy
8. line 239-242: it is not clear why at delivery there were differences in the groups of people studied, and then these differences disappeared despite taking B9.
9. Table 1 is heavily overloaded with material that is poorly perceived.
10. line 468: “800 µg/die” -> “800 µg/day”
11. line 510: “5 mol/L” -> “5 µmol/L”
12. Almost nothing has been said about the effectiveness of lowering homocysteine levels in relation to the prevention of vascular diseases, obesity and kidney diseases in children.
13. Line 675: “odds ratio¼1.5;” what is meant?

Author Response

Reviewer #2

Comment 1

From the abstract it is not clear what role homocysteine is in what pathologies is considered. The summary, in fact, does not contain new, important and interesting points and positive conclusions. It is recommended to make a stronger emphasis on the choice of research objects and the breadth of the studied pathologies.

Answer

We agree with your comment and we have rewritten the abstract as follows:

“Increased plasma homocysteine (Hcy) is a risk factor for several pathological disorders. The present review is focused on the role of Hcy in different population groups especially in risk conditions (pregnancy, infancy, old age), and on its relevance as a marker or etiological factor of the diseases in these age groups focusing on the nutritional treatment of elevated Hcy levels. In pregnancy Hcy levels were investigated in relationship to the increased risk of Adverse Pregnancy Outcomes such as small for gestational age at birth, preenclampsia, recurrent abortions, low birth weight, intrauterine growth restriction. In pediatric population HCy levels are important not only for cardiovascular disease, obesity and renal disease, but the most interesting evidences concern study on elevated levels of Hcy and Autism spectrum disorder (ASD) and Attention Deficit Hyperactivity Disorder (ADHD). Finally, a focus on the principal pathologies of elderly (cardiovascular and neurodegenerative disease, osteoporosis and physical function) is presented. The metabolism of Hcy is influenced B vitamins and homocysteine-lowering vitamin treatments have been proposed. However, clinical trials did not reach consensus about the effectiveness of vitamin supplementation on the reduction in Hcy levels and on an improvement of pathological condition, especially in elderly patients with overt pathologies, suggesting that other dietary and no dietary factors are involved in high Hcy levels. The importance of novel experimental designs focusing on intra-individual variability as a complement to the typical case/control experimental designs and the study of interactions between different factors it should be emphasized.”

Comment 2

Line 15: what does the criterion for the effectiveness of vitamin intake in this context relate to: a decrease in homocysteine levels or clinical indicators?

Answer

We apologize for the inaccuracy; the text has been corrected as follows:

“…consensus about the effectiveness of vitamin supplementation on the reduction in Hcy levels and an improvement of pathological condition…..”

The same correction was reported in the conclusions.

Comment 3

Figure 1: H2S is a by-product of the activity of CBS and CSE and depicting its formation from cysteine as shown in this figure is incorrect. It is also better to designate enzymes, in contrast to metabolites, with a different font or frame for clarity.

Answer

We agree with the reviewer. But, Hcy is a precursor for endogenous hydrogen sulfide generation, our purpose was to simplify the figure but in the same time to highlight the Hcy connections to other metabolic pathways. In fact, recently Cao et al. [2019] reported the Hydrogen Sulfide Synthesis, Metabolism, and Measurement. The endogenous H2S production is derived by four enzymatic pathways as well a small portion via nonenzymatic reduction.

Cao. X.; Ding ,L.;  Xie, Z.Z.; Yang, Y.; Whiteman. M.; Moore, P.K.; Bian, J.S. A Review of Hydrogen Sulfide Synthesis, Metabolism, and Measurement: Is Modulation of Hydrogen Sulfide a Novel Therapeutic for Cancer?. Antioxid Redox Signal. 2019, 31(1):1–38. doi:10.1089/ars.2017.7058

In the figure enzymes were designed with a different font, as suggested.

Comment 4

line 114:  The excess of cellular Hcy (Hcy-SH) should be exported into the bloodstream via an unknown cellular transporter(s).

Answer

Thank you for the comment. We have added the bibliographic references and modified the text as follows:

Several mechanisms  has been proposed involving the removal of the Hcy  to the extracellular compartment . As been reported that an excess of Hcy in EC can be exported to the extracellular medium via transmembrane transport systems Hcy shares carrier systems with cysteine, in a preferable order of alanine-serine-cysteine (ASC)>aspartate and glutamate (XAG)=large branched-chain neutral amino acids (L) transporter systems in vascular endothelial  and ASC>L>XAG in smooth muscle cells. The sodium-dependent system ASC plays a predominant role for Hcy transport in vascular cells. [31].  Moreover, three different aminoacid  transport systems L, A and y+L. Were identified for the transport mechanisms for Hcy across the microvillous plasma membrane of human placenta identified [32]. Probably the specifity of Hcy  transport is due to the ionic composition of the medium. Under the more oxidizing environment found in plasma, Hcy either reacts with protein Cys residues (R-SH) to homocysteinylate these targets (R-S-S-Hcy), or undergoes oxidization to form the disulfide homocystine (Hcy-Hcy) or mixed disulfides with cysteine (Hcy-Cys) and other low-molecular weight thiols (Hcy-SR)

Comment 5

line 118: “Furthermore, the authors [30] hypothesized that homocystine and mixed disulfides derived from plasmatic Hcy oxidation can enter kidney and liver cells via another yet unidentified transporter, where the reducing environment of cells can regenerate Hcy.”

Answer

Thank you for the comment. We have modified the text as follows:

Also, part of homocysteine in a protein-bound form, mainly albumin, enters the cells by endocytosis of these proteins. Sengupta S et al  [33]  proposed that homocysteine can be delivered to cells via its association with albumin,  since albumin is known to bind to at least three endothelial cell membrane proteins  including transcytosed via plasmalemmal vesicles (gp 60), or endocytosed via receptor mediated endocytosis (gp30, gp18).

Comment 6

line 209: “Children with CHD had higher plasma SAM (131 vs. 100 nmol/L) and N,N‐dimethylglycine betaine (DMG)” DMG or betaine?

Answer

DMG

Comment 7

line 205-216: does not explain how these studies are related to the HHcy problem during pregnancy

Answer

We  agree with the  reviewer: Alsayed et al., 2013  do  not reported any  correlation between  HHcy in pregnancy and  Congenital  Hearth Defects in  children. We eliminated this  citation.

As  regards   Do et al., 2019  we explain  the correlation found  between HHCy in  pregnancy and  low birth  weight, adding this  quote in the text: “A stratified analyses according to maternal plasma homocysteine status showed reduction in birth weight in relation to maternal betaine was limited to who had mothers   with HHcy (≥ 5.1 µmol/L)”. 

Comment 8

line 239-242: it is not clear why at delivery there were differences in the groups of people studied, and then these differences disappeared despite taking B9.

Answer

Thank you for the comment. We have modified the text as follows:

No significant difference in serum folate concentration was found between women who use folic acid supplement at week 28 and the nonusers. Serum folate was significantly higher (P < 0.05) in the 86% of women using supplement use than in nonusers at delivery, the median (5th, 95th percentile) values were 9.40 (3.02, 30.92) and 6.24 (1.92, 34.0) nmol/L. The maternal folate concentration was lower than that observed in other studies. The lower folate status observed was most likely related to a low dietary intake which together with suggested low supplement use and lack of folic acid–fortified foods indicates that the total folate intake was less than recommendations. The Authors speculate that “the lack of an association was due to the active transport of these key nutrients across the placenta to maintain fetal status, which attenuated the simple correlations that may be evident in folate replete populations. Folate status was lower and the tHcy concentration was higher than what has been observed in populations in whom supplement use is common and fortified foods form part of the habitual diet; the latter sources provide the vitamin in the synthetic folic acid form, well known to be more bioavailable than the natural food folate forms”

Comment 9

Table 1 is heavily overloaded with material that is poorly perceived.

Answer

Thank you for your suggestion, we have modified the table.

Comment 10

line 468: “800 µg/die” -> “800 µg/day”

Answer

Done. Thank you.

Comment 11

line 510: “5 mol/L” -> “5 µmol/L”

Answer

Done. Thank you.

Comment 12

Almost nothing has been said about the effectiveness of lowering homocysteine levels in relation to the prevention of vascular diseases, obesity and kidney diseases in children.

Answer

Thank you for your  very interesting comment. Unfortunately, there is a lack of  intervention studies  conducted on  children and adolescents demonstrated  the  effectiveness of  lowering   HHcy  in relation to the prevention of cardiovascular diseases and  kidney  diseases. At  present,  studies  demonstrated  the  significance  of HHcy in children an adolescents  as a  independent predictor for increased risk  of  these  diseases  or the effectiveness of  lowering HHcy in reducing  symptoms.

 As  regards obesity  we explain something   in rows 394-399. “Debohkordi et al. [99] in a previous randomized double–blind controlled clinical trial on 60 obese and overweight children aged 5-12 years demonstrated that supplementation of 1mg folic acid per day or 5 mg folic acid per day for 8 week had a positive effect in the reduction of HHcy and insulin resistance”.  Authors, unfortunaltely  do not reported the effect  of lowering HHCy on  body weight, but the reduction of insulin resistence is a good  outcome  for reducing   obesity.

Comment 13

Line 675: “odds ratio¼1.5;” what is meant?

Answer

We apologize for the typo; we have corrected the text as follows: “odds ratio =1.5”

Reviewer 2 Report

Comprehesive review on the population aspects of detrimental role of homocysteine. Notes and comments:1. Last sentence of the abstract should be newly formulated.

Author Response

Reviewer #1:

Comment

Last sentence of the abstract should be newly formulated

Answer

Thank you for your comment. We have rewritten the abstract according to the comment of reviewer 2 and the last sentence was eliminated.

“Increased plasma homocysteine (Hcy) is a risk factor for several pathological disorders. The present review is focused on the role of Hcy in different population groups especially in risk conditions (pregnancy, infancy, old age), and on its relevance as a marker or etiological factor of the diseases in these age groups focusing on the nutritional treatment of elevated Hcy levels. In pregnancy Hcy levels were investigated in relationship to the increased risk of Adverse Pregnancy Outcomes such as small for gestational age at birth, preenclampsia, recurrent abortions, low birth weight, intrauterine growth restriction. In pediatric population HCy levels are important not only for cardiovascular disease, obesity and renal disease, but the most interesting evidences concern study on elevated levels of Hcy and Autism spectrum disorder (ASD) and Attention Deficit Hyperactivity Disorder (ADHD). Finally, a focus on the principal pathologies of elderly (cardiovascular and neurodegenerative disease, osteoporosis and physical function) is presented. The metabolism of Hcy is influenced B vitamins and homocysteine-lowering vitamin treatments have been proposed. However, clinical trials did not reach consensus about the effectiveness of vitamin supplementation on the reduction in Hcy levels and on an improvement of pathological condition, especially in elderly patients with overt pathologies, suggesting that other dietary and no dietary factors are involved in high Hcy levels. The importance of novel experimental designs focusing on intra-individual variability as a complement to the typical case/control experimental designs and the study of interactions between different factors it should be emphasized.”

Round 2

Reviewer 1 Report

1. A number of designations (homocysteine and Hcy) are not unified. Some sections are not divided into paragraphs, which complicates the perception of the text.

Because In the review, the term Hcy by default refers to its general content, then the use of the notation tHcy seems unnecessary.

Some references are not included in the text.

2. Section 2 does not at all pay attention to such an important issue as the role of maternal hyperhomocysteinemia in development of neural tube defects.

3. “line 239-242: it is not clear why at delivery there were differences in the groups of people studied, and then these differences disappeared despite taking B9.

 Answer

 Thank you for the comment. We have modified the text as follows:

 No significant difference in serum folate concentration was found between women who use folic acid supplement at week 28 and the nonusers. Serum folate was significantly higher (P < 0.05) in the 86% of women using supplement use than in nonusers at delivery, the median (5th, 95th percentile) values were 9.40 (3.02, 30.92) and 6.24 (1.92, 34.0) nmol/L. The maternal folate concentration was lower than that observed in other studies. The lower folate status observed was most likely related to a low dietary intake which together with suggested low supplement use and lack of folic acid–fortified foods indicates that the total folate intake was less than recommendations. The Authors speculate that “the lack of an association was due to the active transport of these key nutrients across the placenta to maintain fetal status, which attenuated the simple correlations that may be evident in folate replete populations. Folate status was lower and the tHcy concentration was higher than what has been observed in populations in whom supplement use is common and fortified foods form part of the habitual diet; the latter sources provide the vitamin in the synthetic folic acid form, well known to be more bioavailable than the natural food folate forms”

The authors of the review do not answer this question and therefore it is not clear for what purpose this article ([61]) was included. Does folic acid use affect her levels during pregnancy? According to the source (Wallace et al., [61]), it is generally unclear when their studies were conducted, and data on the “supplement” and “nonusers” groups are not shown, and these groups themselves were formed on the basis of surveys. I do not recommend including this material in the review.

4. Line 204: Pregnant women with preeclampsia had significantly (P ≤0.0001) higher Hcy level (9.8±3.3 μmol/L) than controls (8.4±1.9 μmol/L).

Does the authors think that the values of 9.8 ± 3.3 and 8.4 ± 1.9 μM do not differ so much, even if it is stated that P ≤0.0001? If Hcy levels in these groups really differed significantly, can this indicator be used to diagnose preeclampsia and with what sensitivity and specificity?

doi on ref. [52] is incorrect

5. Line 205: On the other hand, Dodds et al, [53] demonstrated no significant correlation between Hcy levels in early pregnancy and risk of small for gestational age at birth, and any correlation with folic acid supplement intake in the first and second trimesters of pregnancy.

Why do the authors contrast the results of the study of preeclampsia [52] with the results of the study [53] related to small for gestational age, but do not mention the results of the study [53] related to preeclampsia? What is the relationship between these sentences, if they relate to different pathologies?

I did not find in article [53] any mention of “correlation with folic acid supplement intake in the first and second trimesters of pregnancy”

6. Line 213: Hcy were not significantly different among any of the trimesters…

Line 216: Revisiting the role of first trimester homocysteine levels with the maternal and fetal outcome, could be argued that in pregnant women Hcy levels decrease during the first and the second trimester and increase in the second half of the third trimester of pregnancy.

The authors are not confused by this contradiction in the discussion of [54]?

7. Line 229: “with HHcy (≥ 5.1 μmol/L)”

Is it possible to talk about HHcy with a Hcy level of 5.1 μM? Especially since earlier authors write “low levels of Hcy (5.71 mkM ...), line 208. In addition, the study [56] is preliminary.

8. Line 231-243. The authors point to the association of an elevated level of Hcy with pregnancy disorders and a decreased level of B9 with pregnancy disorders. The questions about the effect of consumption of B9 on these disorders remain insufficiently illuminated, and whether the protective effects of B9 are mediated through a decrease in the level of Hcy in pregnancy pathologies.

9. Line 341: Table 1 is still cumbersome and hard to read. This, in essence, is a compilation of many studies performed by different methods, without attempting to generalize and analyze.

10. Line 354: “POTS is a common functional cardiovascular disease in children and adolescents representing one of the main causes of syncope in pediatric population, and it accounts for about 32.2% of all cases [89.90]. POTS syncope is more common in adolescent females and often occurs after menarche [91]. The symptoms of POTS are: nausea, headache, palpitations, dizziness and chest discomfort, blurred vision shortness of breath and, in severe cases, syncope. ”

What is POST by definition?

Line 373: “POT” -> “POTS”?

11. Line 373: “their measurement”

The measurement of what is meant, homocysteine?

12. Line 380:“Moreover, high levels of Hcy correlated to levels of highly sensitive cardiac troponin T (hscTnT) a pivotal biomarker in HF.”

It should be written in more detail about the possibilities of using Hcy as a prognostic marker of HF. Does Hcy level predict HF onset, severity or outcome?

13. Line 390: “CI: 1.10‐1.18” -> “CI: 2.10‐2.18”?

14. Line 398: “However, with regards to Hcy…”

Need a new paragraph

15. Line 455,456: “mol/L” -> “mmol/L”

16. Line 488: “in aetiology of ASD”

Probably we are talking about pathogenesis, and not about etiology

17. Line 580: Need a new paragraph

18. Line 607: It is also interesting to note that folic acid supplementation was significantly more effective in individuals with the MTHFR TT genotype without HHcy (<12.8 mkM) compared with individuals with HHcy in the same genotype.

Author Response

Comment 1

A number of designations (homocysteine and Hcy) are not unified. Some sections are not divided into paragraphs, which complicates the perception of the text.

Because In the review, the term Hcy by default refers to its general content, then the use of the notation tHcy seems unnecessary.

Some references are not included in the text.

Answer

Thank you for your suggestion.

We unified some designations ( homocysteine and Hcy) as suggested and  we  substitute tHcy with  Hcy when necessary.

We added the missing references in the text.

Comment 2

Section 2 does not at all pay attention to such an important issue as the role of maternal hyperhomocysteinemia in development of neural tube defects.

Answer

We added some studies in the section 2 related to maternal homocysteine and the development of neural tube defects.

Significance of HHcy levels ……..  and Neural Tube Defects [49-51] ……..

 ………. As regards Neural Tube Defects low vitamin B12 and B9  status are  risk factors for  Neural  Tube Defects (NTDs) ………………………………………………………………. of Zhang et al. [65],  that suggested that higher levels of Hcy contribute  to NTDs through up-regulation of histone H3K79Hcy, leading to abnormal expressions of selected NTC-related genes.

Comment 3

line 239-242: it is not clear why at delivery there were differences in the groups of people studied, and then these differences disappeared despite taking B9.

 Thank you for the comment. We have modified the text as follows:

 No significant difference in serum folate concentration was found between women who use folic acid supplement at week 28 and the nonusers. Serum folate was significantly higher (P < 0.05) in the 86% of women using supplement use than in nonusers at delivery, the median (5th, 95th percentile) values were 9.40 (3.02, 30.92) and 6.24 (1.92, 34.0) nmol/L. The maternal folate concentration was lower than that observed in other studies. The lower folate status observed was most likely related to a low dietary intake which together with suggested low supplement use and lack of folic acid–fortified foods indicates that the total folate intake was less than recommendations. The Authors speculate that “the lack of an association was due to the active transport of these key nutrients across the placenta to maintain fetal status, which attenuated the simple correlations that may be evident in folate replete populations. Folate status was lower and the tHcy concentration was higher than what has been observed in populations in whom supplement use is common and fortified foods form part of the habitual diet; the latter sources provide the vitamin in the synthetic folic acid form, well known to be more bioavailable than the natural food folate forms”

The authors of the review do not answer this question and therefore it is not clear for what purpose this article ([61]) was included. Does folic acid use affect her levels during pregnancy? According to the source (Wallace et al., [61]), it is generally unclear when their studies were conducted, and data on the “supplement” and “nonusers” groups are not shown, and these groups themselves were formed on the basis of surveys. I do not recommend including this material in the review.

Answer

Thank you, we agree with your consideration and we eliminated the study (61) from the review

Comment 4

Line 204: Pregnant women with preeclampsia had significantly (P ≤0.0001) higher Hcy level (9.8±3.3 μmol/L) than controls (8.4±1.9 μmol/L).

Does the authors think that the values of 9.8 ± 3.3 and 8.4 ± 1.9 μM do not differ so much, even if it is stated that P ≤0.0001? If Hcy levels in these groups really differed significantly, can this indicator be used to diagnose preeclampsia and with what sensitivity and specificity?

doi on ref. [52] is incorrect

Answer

We agree with your comment and we deleted this citation. Furthermore,  the study  concerns  few patients and it’s older than   other  studies  included  in  our review ( 51 and  60). In our opinion  HHcy  can not be used for diagnose of preeclampsia but as  indicator of  its increased risk when the   its levels are above 9.0.  

Comment 5

Line 205: On the other hand, Dodds et al, [53] demonstrated no significant correlation between Hcy levels in early pregnancy and risk of small for gestational age at birth, and any correlation with folic acid supplement intake in the first and second trimesters of pregnancy.

Why do the authors contrast the results of the study of preeclampsia [52] with the results of the study [53] related to small for gestational age, but do not mention the results of the study [53] related to preeclampsia? What is the relationship between these sentences, if they relate to different pathologies?

I did not find in article [53] any mention of “correlation with folic acid supplement intake in the first and second trimesters of pregnancy”

Answer

According to your suggestion, we modified the sentences as:

In a previous study, Dodds et al, [53] demonstrated no significant correlation between Hcy levels in early pregnancy and risk of small for gestational age at birth, meanwhile positive correlation between Hcy concentration  at the top decile and pregnancy loss and preeclampsia was found (RR 2.1, 95% CI 1.2–3.6 and  RR 2.7, 95% CI 1.4 –5.0, respectively). The study do not observed a dose response relationship according to quartile of Hcy concentration, due to the small sample size.

Comment 6

Line 213: Hcy were not significantly different among any of the trimesters…

Line 216: Revisiting the role of first trimester homocysteine levels with the maternal and fetal outcome, could be argued that in pregnant women Hcy levels decrease during the first and the second trimester and increase in the second half of the third trimester of pregnancy.

The authors are not confused by this contradiction in the discussion of [54]?

Answer

We agreed with your comment. The sentence at line 216  is relative to the  study of   Mascarenhas et al. [55], and  then we substituted  the sentence in line 216 as:

Mascarenhas et al. [56], suggested that the evaluation of Hcy levels during pregnancy should be done between 8 to 12 weeks  to  have  a good indicator  for pregnancy outcome. Increased first trimester of pregnancy serum Hcy is associated with history of pregnancy losses, hypertensive disorders of pregnancy, and preterm birth. This is also associated with hypertensive disorders of pregnancy, pregnancy loss, oligohydramnios, meconium stained amniotic fluid, and low birth weight in the current pregnancy

Comment 7

Line 229: “with HHcy (≥ 5.1 μmol/L)”

Is it possible to talk about HHcy with a Hcy level of 5.1 μM? Especially since earlier authors write “low levels of Hcy (5.71 mkM ...), line 208. In addition, the study [56] is preliminary.

Answer

We agree with your comment and we deleted the study form the review

Comment 8

Line 231-243. The authors point to the association of an elevated level of Hcy with pregnancy disorders and a decreased level of B9 with pregnancy disorders. The questions about the effect of consumption of B9 on these disorders remain insufficiently illuminated, and whether the protective effects of B9 are mediated through a decrease in the level of Hcy in pregnancy pathologies.

Answer

Your consideration is very pertinent and the cited studies (57, 58,59)  do not referred  to a decrease of homocysteine in relantionship to  an increased folate status decreasing the risk of pregnancy disorder .  We  decided  to delete  these sentences and relative references.  We added as follows:

Hcy levels during pregnancy are influenced both by vitamin B nutritional status of women and by some specific physiological factors including increased glomerular filtration, rate hemodilution and endocrinological changes occurring in this period, leading in a normal condition to a reduction of Hcy in the midpregnancy period [57]. Moreover Gaiday et al. [52] concluded that Hcy levels during pregnancy seem linked  also to geographical, cultural and social characteristics of population

Comment 9

Line 341: Table 1 is still cumbersome and hard to read. This, in essence, is a compilation of many studies performed by different methods, without attempting to generalize and analyze.

Answer

As your requested, we simplified Table 1. improving the text as follows:

Bailey et al., [66] defined cut off levels for Hcy in healthy children and adolescents in the Canadian Laboratory Initiative for Pediatric Reference Intervals (CALIPER) program.  ………………………………………………………………………………………………………………………………………………………….. Demand for creatine biosynthesis in men is usually greater than in woman, and the link between creatine biosynthesis and Hcy metabolism is the enzyme guanidinoacetate methyltransferase [75].

Comment 10

Line 354: “POTS is a common functional cardiovascular disease in children and adolescents representing one of the main causes of syncope in pediatric population, and it accounts for about 32.2% of all cases [89.90]. POTS syncope is more common in adolescent females and often occurs after menarche [91]. The symptoms of POTS are: nausea, headache, palpitations, dizziness and chest discomfort, blurred vision shortness of breath and, in severe cases, syncope. ”

What is POST by definition?

Line 373: “POT” -> “POTS”?

Answer

We are sorry for Typo of POTS. The correct definition is  Postural Tachycardia Syndrome (PoTS). We have replaced the term across the text.

Comment 11

Line 373: “their measurement”

The measurement of what is meant, homocysteine?

Answer

We substituted the term with “the measurement of Hcy”

Comment 12

Line 380:“Moreover, high levels of Hcy correlated to levels of highly sensitive cardiac troponin T (hs‐cTnT) a pivotal biomarker in HF.”

It should be written in more detail about the possibilities of using Hcy as a prognostic marker of HF. Does Hcy level predict HF onset, severity or outcome?

Answer

Thank you for your suggestion. We specified better the main findings of the Authors in the text as follows:

In patients with CHF the authors suggested cut-off point for Hcy>8.1 µmol/L and Hs-cTnT>10 pg/mL levels as diagnostic predictor. The Authors reported both plasma Hcy and serum hs-cTnT levels significantly (P<0.001) increased levels in adverse outcome (Hcy 13.23 ± 1.03 µmol/L; Hs-cTnT 110.50 ± 10.98 pg/ml) with respect to the favorable outcome (10.96 ± 0.87 µmol/L and 69.78 ± 16.35 pg/ml), respectively. In addition, a significant increase of the mean levels of hs-cTnT and Hcy were associated with severity of HF (Class IV patients vs Class III and Class II patients (p = 0.001).

Comment 13

Line 390: “CI: 1.10‐1.18” -> “CI: 2.10‐2.18”?

Answer

We reported the CI value as in the manuscript.

Comment 14

Line 398: “However, with regards to Hcy…”

Need a new paragraph

Answer

Done

Comment 15

Line 455,456: “mol/L” -> “mmol/L”

Answer

Thank you. We have corrected with μmol /L

Comment 16

Line 488: “in aetiology of ASD”

Probably we are talking about pathogenesis, and not about etiology

Answer

Done

Comment 17

Line 580: Need a new paragraph

Answer

Done

Comment 18

Line 607: It is also interesting to note that folic acid supplementation was significantly more effective in individuals with the MTHFR TT genotype without HHcy (<12.8 mkM) compared with individuals with HHcy in the same genotype

It is also interesting to note that folic acid supplementation was significantly more effective in individuals with the MTHFR TT genotype without HHcy (<12.8 mkM ) compared with individuals with HHcy in the same genotype

Answer

Thank you. We have added the comment in the text:

It is also interesting to note that folic acid supplementation was significantly more effective in individuals with the MTHFR TT genotype without HHcy (<12.8 mmol/L) compared with individuals with HHcy in the same genotype. The authors speculate that differences in folate status, in MTHFR enzyme activity  such as long-standing injury between participants with CC/CT genotype and those with TT genotype may be responsible for the observed results and speculate that higher doses of folic acid or longer supplementation periods would be needed.

Round 3

Reviewer 1 Report

The new version of the manuscript has been improved, but the authors need to carefully review the text for minor typos, for example

Line 241:Defects (NTDs) determining an increase of Hcy levels ( [59‐61].
Line 243: Deb et coll [62] observed that the mean homocysteine Hcy levels …
Line 261: “up‐regulation of histone H3K79Hcy “ -> “up‐regulation of histone H3 N-homocysteinylation at Lys-79”

This manuscript is a resubmission of an earlier submission. The following is a list of the peer review reports and author responses from that submission.

Round 1

Reviewer 1 Report

This mini review is dedicated to the role of homocysteine (Hcy) in various pathologies occurring at an early and old age. As is common the manuscript is divided into three parts (Hcy metabolism, hyperhomocysteinemia (HHcy) in children, HHcy in the elderly) and conclusion. The issue of HHcy remains relevant in preventing and treating of various diseases. A lot of reviews have been published on this topic in the recent years.

Unfortunately, I was not quite satisfied by the quality of the material and cannot find sufficient reasons for which this review can be of interest to the readers. This review does not provide clear proposals on the prospects for solving the problem under discussion. Also the authors did not carry out an in-depth literature analysis. Therefore, I cannot recommend this manuscript for publication.

Major comments

The information on Hcy metabolism is poorly used for the subsequent      analysis of clinical studies.

Sections 2 and 3 are independent of each other and do not relate to      the introduction and the conclusion. What are the features of Hcy      metabolism in children and the elderly? What is Hcy relevance as a marker      or etiological factor of the diseases in these age groups?

The review does not address the problem of HHcy in pregnancy and      gynecology at all and very little attention is paid to the priority areas:      the connection of HHcy with complications of cardiovascular diseases      (stroke, infarction, coronary heart disease, etc.) and the latest data on      the effectiveness of Hcy-lowering therapy.

The issue of the search for alternative markers among Hcy      metabolites almost was not discussed.

The conclusions are not very clear. What do the authors mean by “intra-individual      variability”? How can it help solve the problems mentioned in the      Conclusion?

Minor comments

Line 16: It is not clear what is meant by “functional decline”

Line 69: Unlike in rats, the role of kidneys in the reabsorption of plasma Hcy is not completely clear in humans.

Line 76: Fig. 1 should be narrowed down so that the structures are not distorted. Also, the figure does not present the protein-bound form of Hcy.

Line 97: The authors mention Hcy thiolactone but do not explain where it comes from and what it is.

Line 98 and on:  Do the authors mean hyperhomocysteinemia by “hyper-Hcy”? If yes, it is better to use a commonly used abbreviation HHcy.

Line 112: blood plasma Hcy

Line 133: 2. Children and Adolescents

Line 146: “increasingly prevalent aetiology” is a not clear phrase

Line 171:  By “mcg/die” do the authors mean mkg/day?

Line 229: What is meant by “found patients with HF with Hcy plasma levels before treatment”? Adults and children with HF? What is meant by “treatment”?

Line 234: HCY -> Hcy

Line 254 and 266: hypermocisteinemia -> hyperhomocysteinemia

Line 273: 3. Elderly

Line 358: e coll -> et al.? Judging by the title, the reference 102 is dedicated to Alzheimer’s disease and not to bone fractures in the context of which it is given.

Reviewer 2 Report

Paper presents superficial overview of the extracted literature data on the potentially detrimental effect of hHcy in the different age lifespan disorders. Several papers in renowned journals summarized and discussed the role of Hcy in the age associated disorders (see Ageing Res Rev 49, 144, 2019 and others) in a detailed way.

Submitted paper presents only quite superficial overview of different disorders with the relation to the level of Hcy, howver missing clinico anatomical presentations, modality and heterogeneity of the diseases with the corresponding clinical manifestations. Authors did not discuss asymptomatic prodromal and symptomatic features of the given disorders and in general etiopathogenetic aspects is not presented in a valueable level.The novelty of the study is questionable, even unifying selection of the disorders is Hcy.

Reviewer 3 Report

This manuscript tries to describe the role of homocysteine at different age of lifespan. The review has many flaws that make the interpretations and conclusions difficult to accept. For starters, lack of data of others segments populations such as adults is also of concern. 

Figure 1. the legend are missing from the figure.

There are previous papers, investigating the effects of hyperhomocysteine in several pathologies such as gastrointestinal disorders, cancer, hearing loss…, that should be cited in the introduction: I do think citing the wider literature is important in strengthening the rationale for this paper.

The authors described methionine and folate metabolism and connecting pathways. They should cite some paper...

Authors need to discuss the mechanism in all cases. For example, flotae is cofactor in homocysteine metabolism. Folate deificency increase homocysteine levels, inducing apoptosis of neurons (Wang J et al., 2012; Akchiche N et al., 2012; Herrmann & Obeid, 2011; Moore P et al., 2001; Kruman IIet al., 2000).

There are abbreviatures and genes nomenclature that are incorrect. For example, hyperhomocysteinemia is HHcy not hyperHcymia or methylenetetrahydrofolate reductase (MTHFR) should write in italic and the nomenclatural systems also provide for at least human-versus-nonhuman specificity by using different capitalization such as MTFHR for human and Mtfhf for animals